# Ra-NEM: Faithful model explanations through stochastic feature selection

## Abstract

The adoption of machine learning for socially relevant tasks requires effective explainable artificial intelligence (XAI) methods to better understand the behavior of machine learning models. Attribution methods are a popular XAI approach in which input-output relationships are characterized by heat maps that reflect the relative importance of input features for a particular prediction. The quality of such maps is often assessed by measuring faithfulness based on the area under the insertion curve. We propose the first method that directly optimizes this metric to generate attribution heat maps. We establish the connection between insertion curves and top-$k$ feature selection, which leads to a loss function measuring the quality of attributions. Randomization of the loss allows us to efficiently approximate its gradient. We combine the loss function with the neural explanation mask framework to create a new approach for providing accurate attributions efficiently. Experiments demonstrate superior faithfulness along with robust attributions and low inference time, suggesting a new path to generate useful explanations. Code is available at: https://anonymous.4open.science/r/Ra-nem_ICLR-2AD4

## 1 Introduction

Explainable AI (XAI) aims to clarify how and why a model produces specific outputs from given inputs. This insight is needed to provide a higher level of transparency and safety in real-world applications (Gerlings et al., 2021) and might soon be a regulatory mandate (Chung et al., 2024). There are many different tasks under the umbrella term XAI. One of the most fundamental ones is to estimate the importance of individual input features with respect to model output, usually through a feature map, also referred to as an *attribution*, which assigns to each input feature some importance score. Although verification of attribution maps is still an open research question, one quality commonly measured during evaluation is *faithfulness*, which attempts to quantify how well the attribution aligns with the model's behavior for a given input-output pair. Faithfulness, in turn, is often assessed through *deletion* and *insertion curves* (Petsiuk et al., 2018; Muzellec et al., 2024; Fong and Vedaldi, 2017). Introduced by Petsiuk et al. (2018), the insertion curve measures changes in the model output as features are added in order of highest importance. A higher area under the curve (AUC) indicates better explanations. Similarly, the deletion curve tracks how much the output changes when the features are removed in reverse order. In this case, a low AUC indicates a good attribution. Although many different heuristics have been proposed to solve the attribution problem (Abhishek and Kamath, 2022), none, to our knowledge, has approached it by optimizing the insertion and/or deletion curve directly. Assuming that faithfulness defined based on these curves is a preferable quality for an attribution, then directly optimizing the area under these curves would be a principled way of generating high quality attributions. We present a method that does exactly that, generating attributions that directly optimize the AUCs while being fast enough for real-time applications.

This study **(i)** establishes a link between top-$k$ feature selection with variable $k$ and the insertion and deletion curves; **(ii)** states the sample complexity of Monte Carlo approximations of the area under these curves; **(iii)** proposes a differentiable operator for feature masking depending on the top-$k$ features selected via the "Gumbel-top trick" (Kool et al., 2019); **(iv)** adds the new operator to the Neural Explanation Mask (NEM) framework (Møller et al., 2024) yielding *Ranking NEM* (Ra-NEM), which produce sparse faithful explanations with low latency; and **(v)** experimentally evaluates Ra-NEM applied to different deep learning architectures for image classification in comparison to state-of-the-art attribution methods, demonstrating that the new approach performs on average not

only superior in terms of faithfulness but also when looking at sparseness and complexity while inheriting the speed of the NEM approach.

## 2 RELATED WORK

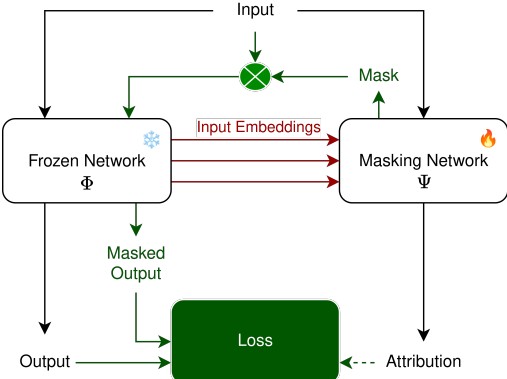

Figure 1: The general Neural Explanation Method (NEM) framework's operation during inference (black and red arrows) and training (black, red, and green arrows). During inference, the input is processed by the frozen network, which generates its output, and the masking network, which produces an attribution (explanation). Depending on the NEM architecture, representations from the frozen network may assist the masking network in generating the attribution. During training, the masking network generates a mask from the attribution. In Møller et al. (2024; 2025), this mapping is the identity, and in this work it is a differentiable top-$k$ feature selector. The mask is applied to the input and passed through the frozen network to produce a masked output. Both masked and unmasked outputs are used in a loss function to optimize the masking network. Prior work has also incorporated the attribution into the loss.

Attribution of model input-output relations is a well-established and growing field. Many heuristics have been proposed to solve the attribution problem for different modalities such as images, texts, graphs, etc. (Yang et al., 2023). The approaches, especially in the context of computer vision, can be roughly divided into two families, occlusion- and gradient-based methods (Fong and Vedaldi, 2017). The latter leverages the gradient of the given model w.r.t. the input to determine the sensitivity of the output to the individual input features. These methods include Saliency (Simonyan et al., 2014), GradSHAP (Lundberg and Lee, 2017), GradCAM++ (Chattopadhay et al., 2018), and Integrated Gradients (Sundararajan et al., 2017). The low latency of such methods usually comes with low faithfulness, especially for some specific neural architectures such as CNNs (Muzellec et al., 2024). The occlusion-based methods introduce perturbation/occlusion to the input and measure the changes in the output to determine the importance of various features. Examples include RISE (Petsiuk et al., 2018), Information Bottleneck Attribution (IBA, Schulz et al., 2020), and Smooth pixel mask (Fong et al., 2019). Generally, occlusion-based methods exhibit good faithfulness at the cost of slower inference speed. A notable exception to this rule is the NEMt method (Møller et al., 2025), an instantiation of the Neural Explanation Mask (NEM) framework (Møller et al., 2024), which achieves occlusion-based masking at low latency by predicting a mask directly instead of running a costly optimization process for each input as done by other occlusion-based methods. XAI methods can also be categorized according to whether they generate additive feature explanations or set-of-feature explanations (Fong et al., 2019; Møller et al., 2024). Additive feature explanations provide attributions which rank individual features based on their importance. Examples of such methods include GradCAM (Chattopadhay et al., 2018), RISE (Petsiuk et al., 2018), and Integrated Gradients (Sundararajan et al., 2017). In contrast, set-of-feature explanations identify the minimal subset of features that produce a similar output to the full feature set. Unlike additive explanations, these methods do not rank every feature, making them less suitable for evaluation with deletion and insertion curves. Examples include Smooth Pixel Mask (Fong et al., 2019), NEMt (Møller et al., 2025), and Learn2Explain (LX2, Chen et al., 2018). L2X uses top-$k$ feature selection, employing the

Gumbel softmax operator iteratively to identify up to $k$ important features, where $k$ ranges from 4 to 10 in the original paper. To our knowledge, the framework has not been tested for larger values of $k$.

**Neural Explanation Masks.** Our approach extends the *Neural Explanation Mask* (NEM) framework (Møller et al., 2024). The idea of NEM is to augment a given differentiable model $\Phi$ with an explanation module $\Psi$ that generates a feature attribution $a = \Psi(x)$ for the model output $y = \Phi(x)$; see Figure 1 for an overview of the NEM framework. The explanation module $\Psi$ is trained on a corpus of (unlabelled) data. When deployed, providing an explanation corresponds to (simultaneously) evaluating $\Psi$, which leads to low latency. The models $\Phi$ and $\Psi$ are not optimized in tandem, which is why $\Phi$ is also called the *frozen model*. This is in contrast to other occlusion-based methods, which usually need to perform a separate optimization process for each new input to generate an explanation. Different NEM architectures (Møller et al., 2024; 2025) can be derived depending on the architecture $\Psi$ and how it is integrated into the explained model $\Phi$. Previous work has focused on the NEM-U (NEM using U-Net structure, Ronneberger et al., 2015) architecture, where $\Phi$ and $\Psi$ are seen as the encoder and decoder of a U-Net architecture, respectively, meaning that $\Psi$ utilizes skip connections to extract intermediate input representations from $\Phi$. It is appealing that the NEM framework does not interfere with training and using $\Phi$, the main modeling task. However, one hindrance to the adoption of all current NEM methods is the need to choose hyperparameters that specify the trade-off between accuracy and complexity. Furthermore, all previous NEM methods produce a set of important features so they can only determine whether a feature is important or not, but not how important it is compared to others. Thus, no previous NEM methods can be used to reliably determine the $k$ most important features for a given $k$.

## 3 METHODOLOGY

This section presents our theoretical results and derives an XAI algorithm based on these results. First, we establish the relationship between the insertion and deletion curves and top-$k$ feature selection. This discussion lends itself to an objective function for learning feature attributions. To make the objective tractable, we use Monte Carlo approximation, and to make it differentiable, we employ the "Gumbel top-$k$ trick" for sampling without replacement. Then we show how the Neural Explanation Mask framework can be extended to our new *Ranking-NEM* (Ra-NEM) architecture using the derived objective function.

### 3.1 APPROXIMATING THE INSERTION CURVE VIA TOP-$k$ FEATURE SELECTION

The basic idea is to generate feature attributions by optimizing the AUC of the corresponding insertion (and/or deletion) curve to maximize the faithfulness of the attribution directly. Let $x \in \mathbb{R}^d$ be some $d$-dimensional input (e.g., a flattened image) and $\Phi : \mathbb{R}^d \to \mathbb{R}^c$ some fixed model with $c$-dimensional output for which we want to provide explanations. This $c$-dimensional output could be the logits for $c$ classes but also any other representation learned by $\Phi$, in particular some hidden layer embedding in a deep neural network. The set of indices corresponding to the $d$ input features is denoted by $[d] = \{1, \ldots, d\}$. For a given input $x$, a *feature attribution* is a vector $a \in \mathbb{R}^d$, where $a_i < a_j$ indicates that the feature $i$ is considered more relevant than the feature $j$. Each feature attribution $a$ defines the corresponding *feature ranking* $\sigma$. We describe the ranking by the permutation $\sigma_a : [d] \to [d]$, which orders the features in descending order breaking ties according to some deterministic rule. That is, $\sigma_a(i) < \sigma_a(j)$ iff $a_i < a_j$.

**Insertion and deletion curves.** Let $\omega : \mathbb{R}^d \times \mathcal{P}([d]) \to \mathbb{R}^d$, where $\mathcal{P}$ denotes the power set, be some operator responsible for perturbing/masking features in $x$. The second argument specifies the features that should *not* be perturbed. For example, the second argument could specify all pixels in an image $x$ that are not replaced by a constant when masking $x$.[1] We define a function $\delta_\Phi : \mathbb{R}^d \times \mathbb{R}^d \to [0, 1]$ that measures the quality of the perturbed input with respect to the original input and the model. This could be the logit of the predicted class (Petsiuk et al., 2018) or, in an unsupervised setting, the similarity of latent space representations (Wickstrøm et al., 2023; Møller et al., 2024).

---

[1] In this study, we assume fixed length input vectors. However, the approach can also be adopted for variable length inputs. For example, the second argument could specify the set of elements (tokens) to be removed from an input sequence $x$.

The insertion curve is constructed by iteratively adding features to the empty set of features in the order of increasing importance according to some attribution and measuring the difference between the model output of the reduced feature set and the full feature set. We define the *insertion curve* value at a given number of features $i$ by

$$c_{\text{ins}}(i, x, \sigma_a) = \delta_\Phi(x, \omega(x, \{\sigma_a^{-1}(j) \,|\, 1 \le j \le i\})) \ , \tag{1}$$

That is, $c_{\text{ins}}(i, x, \sigma_a)$ measures the similarity of the outputs of $\Phi$ given $x$ and the results of adding the most important features according to feature attribution $a$. The corresponding AUC (area under the curve) is given by

$$\text{AUC}_{\text{ins}}(x, a) = \frac{1}{d} \sum_{i=1}^{d} c_{\text{ins}}(i, x, \sigma_a) \ . \tag{2}$$

We define the complementary *deletion curve* resulting from iteratively removing features in the order of decreasing relevance according to some attribution as

$$c_{\text{del}}(i, x, \sigma_a) = 1 - \delta_\Phi(x, \omega(x, \{\sigma_a^{-1}(j) \,|\, 1 \le j \le d - i\})) \tag{3}$$

and $\text{AUC}_{\text{del}}$ accordingly. From $c_{\text{ins}}(i, x, \sigma_a) = 1 - c_{\text{del}}(d - i, x, \sigma_a)$ it follows that $\text{AUC}_{\text{ins}}(x, a) = 1 - \text{AUC}_{\text{del}}(x, a)$ if the same $\omega$ is used for both curves. However, typically different $\omega$ operators are used, denoted by $\omega_{\text{del}}$ and $\omega_{\text{ins}}$ in the following. Petsiuk et al. (2018) suggest to use $\omega_{\text{del}}(x, T)_i = 0$ and $\omega_{\text{ins}}(x, T)_i = \text{blur}(x)_i$ for $i \in T \subseteq [d]$, where $\text{blur}$ is a Gaussian smoothing operator (we refer to Petsiuk et al., 2018, for a discussion). We follow this standard when computing *faithfulness*. A third option is random replacement $\omega_{\text{rnd}}(x, T)_i \sim \mathcal{N}(x)$ for $i \in T$, where $\mathcal{N}(x)$ is a Gaussian distribution with mean and variance matching the input statistics.

Using the insertion curve criterion, the optimal feature ranking is the permutation $\sigma^*$

$$\sigma^* = \arg\max_{\sigma \in \mathcal{S}_d} \sum_{i=1}^{d} c_{\text{ins}}(i, x, \sigma) \ , \tag{4}$$

where $\mathcal{S}_d$ is the set of all permutations of $[d]$, the positive integers up to $d$. To keep the notation simple, we assume that all $\arg\max$ operations return a unique element. However, the following considerations also hold if a set of equivalent solutions is returned. They also hold when optimizing the area under the deletion curve or a combination $\text{AUC}_{\text{ins}}(x, a) - \text{AUC}_{\text{del}}(x, a)$ (with different choices of $\omega$), which follows from the equivalence of deletion and insertion curves derived above.

The question arises under which conditions $\sigma^*$ defines the optimal subset of $n$ features in terms of $\delta_\Phi$ for any $n \in [d]$. Let the set of the $n$ most important features be defined as

$$T_n^* = \arg\max_{T \in \mathcal{P}(\{1, \ldots, d\}) \wedge |T| = n} \delta_\Phi(x, \omega(x, T)) \ . \tag{5}$$

Then *monotonicity in the set of important features* is defined as the property $T_n^* \subset T_{n+1}^*$. This monotonicity is an implication of a common more restrictive assumption in the attribution literature called *additive feature attribution* (Lundberg and Lee, 2017), which states that any given model output can be approximated by a linear combination of input features.

Assuming monotonicity, there exists a permutation $\sigma_n$ with $\{\sigma_n^{-1}(i) \,|\, i = 1, \ldots, n\} = T_n^*$. Definition (5) implies $\delta_\Phi(x, \omega(x, \{\sigma^{-1}(j) \,|\, 1 \le j \le n\})) \le \delta_\Phi(x, \omega(x, T_n^*))$ for all permutations $\sigma \in \mathcal{S}_d$. Thus, $\sigma_n$ maximizes (2), that is, $\sigma^* = \sigma_n$. Therefore, under the monotonicity assumption, we have $T_n^* = \{\sigma^{*-1}(i) \,|\, i = 1, \ldots, n\}$, that is, optimizing the AUC gives the optimal subset of features for any subset size.

**Monte Carlo approximation.** The optimization problem defined by (4) is typically very high-dimensional in practice, for example, in computer vision tasks where $d$ is the number of pixels in an image. Thus, evaluating the insertion curve at all points during optimization becomes infeasible. Therefore, we consider the Monte Carlo approximation of $\text{AUC}_{\text{ins}}$ given by

$$\widetilde{\text{AUC}}_{\text{ins}}(x, a) = \frac{1}{|\mathcal{J}|} \sum_{i \in \mathcal{J}} c_{\text{ins}}(i, x, \sigma_a) \ , \tag{6}$$

where $\mathcal{J} \subseteq [d]$ is drawn uniformly at random. We have $\mathbb{E}[\widetilde{\text{AUC}}_{\text{ins}}(x, a)] = \text{AUC}_{\text{ins}}(x, a)$, where the expectation is over draws of $\mathcal{J}$, and accordingly $\sigma^* = \arg\max_{\sigma \in \mathcal{S}_d} \mathbb{E}\left[\widetilde{\text{AUC}}_{\text{ins}}(x, a)\right]$. We provide a bound for the sample complexity of the Monte Carlo approximation in Appendix A.

## 3.2 A DIFFERENTIABLE LOSS IN TERMS OF TOP-$k$ FEATURE SELECTION

The first (top) $k$ elements of a feature attribution $a$ are given by $\text{top}_k(a) = \{\sigma_a^{-1}(i) \mid i = 1, \dots, k\}$, and we use the same notation for the $[k] \to [d]$ mapping $\text{top}_k(a)(i) = \sigma_a(i)$. We can rewrite (6) as

$$\widetilde{\text{AUC}}_{\text{ins}}(x, a) = \frac{1}{|\mathcal{J}|} \sum_{i \in \mathcal{J}} c_{\text{ins}}(i, x, \text{top}_i(a)) \ . \tag{7}$$

Instead of optimizing over the space of rankings, we consider the real-valued optimization problem

$$a^* = \arg\max_{a \in \mathbb{R}^d} \widetilde{\text{AUC}}_{\text{ins}}(x, a) \ . \tag{8}$$

Again, we assume that $\arg\max$ returns a single solution to keep the notation simple. We have $\sigma^* = \arg\max_{\sigma \in \mathcal{S}_d} \widetilde{\text{AUC}}_{\text{ins}}(x, a) = \sigma_{a*}$. Plugging in (1), the optimization problem (8) gives rise to the loss function

$$\mathcal{L}(a \mid x) = -\sum_{k \in \mathcal{J}} \delta_\Phi(x, \omega(x, \text{top}_k(a)) \tag{9}$$

to be minimized over $a \in \mathbb{R}^d$ for a training input $x$, where $\mathcal{J}$ is sampled anew in each evaluation of the loss. Thus, instead of improving an attribution by optimizing the AUC of its insertion and/or deletion curve, we can instead optimize top-$k$ feature selection with uniformly sampled $k$ maximizing $\delta_\Phi$.

Optimizing the AUC directly is a non-differentiable problem, but approximate gradients for the rephrased problem can be derived, as will be shown in the following. For gradient-based minimization of (9), we need a differentiable approximation of the top-$k$ selection done by $\omega$. Differentiable top-$k$ feature selection is commonly implemented using the Gumbel softmax operator (Jang et al., 2017; Chen et al., 2018). The Gumbel softmax operator represents a categorical distribution with a continuous and differentiable distribution (Maddison et al., 2017). However, the Gumbel softmax operator is insufficient for top-$k$ feature selection, as it samples only a single element from the categorical distribution. Previous work in the explainability literature (Chen et al., 2018) addresses this limitation by applying the operator $k$ times to the network output, producing a set containing at most different $k$ elements. However, this approach becomes computationally infeasible for large $k$, since the number of operations scales linearly with $k$. Alternatively, top-$k$ feature selection can be achieved by what Kool et al. (2019) refers to as the "Gumbel top-$k$ trick". An ordered sample without replacement from a categorical distribution of size $k$ can be drawn by perturbing the (unnormalized) log-probabilities with values drawn from a standard Gumbel distribution and then selecting the top-$k$ largest elements. Let $a \in \mathbb{R}^d$ be a feature explanation, as, for example, produced by a neural network that provides an explanation for a given input $x$. We interpret $a$ as non-normalized log-probabilities that define the discrete probability distribution $p$ over $[d]$ with

$$p(i) = \frac{\exp(a_i)}{\sum_{j \in [d]} \exp(a_j)} \ . \tag{10}$$

Let $g \in \mathbb{R}^d$ be a random vector drawn from a standard $d$-dimensional Gumbel distribution. Then $\text{top}_k(a + g)$ gives an ordered sample of $k$ elements drawn without replacement from the distribution $p$. For a proof, we refer to Kool et al. (2019). In contrast to iterative applications of the Gumbel softmax, this sampling can be efficiently computed for large $k$ making it suitable for high-dimensional data. Thus, (9) becomes

$$\ell(a \mid x, g, k) = -\delta_\Phi(x, \omega(x, \text{top}_k(a + g)) \tag{11}$$

for a feature attribution $a$ given the input $x$, a number of features $k$, and a sample $g$ from the standard $d$-dimensional Gumbel distribution.

To summarize, by adopting differentiable top-$k$ feature selection via the Gumbel top-$k$ trick, we are able to create an efficient differentiable approximation to $\omega$, which in turn allows us to optimize our derived loss Equation 11. In Appendix E, we discuss the concrete implementation of $\omega$ in more detail, including pseudocode.

## 3.3 RA-NEM

The $\omega$ operator does not have any trainable parameters and simply selects an ordered subset from a given feature attribution in a differentiable way. We can combine it with the NEM-U architecture in a

straight forward way. We feed the explanation produced by $\Psi$ into Algorithm 1 (see Appendix E). Then we can train $\Psi$ end-to-end by gradient-based optimization of (11). That is, for each input $x$ during training, we compute $\frac{\partial}{\partial w} \ell(\Psi(x) \mid x, g, k)$ for randomly drawn $g$ and $k$, where $w$ denotes the parameters (weights) of $\Psi$. We refer to this new approach as *Ranking NEM* (Ra-NEM), because it ranks all input features via the $\Psi$ output processed by Algorithm 1, which stand in contrast to previous NEM variants.Ra-NEM applies binary masks during training (i.e., M in Algorithm 1 is in $\{0, 1\}^d$), while previous NEM methods work with continuous masks $m$, typically $m = a$ directly, and evaluate $\Phi$ on the element-wise product $x \odot m$. This is a crucial difference. While $x \odot m$ may be a valid input when processing images and other fixed-length continuous signals, this is not necessarily the case for other modalities, such as sequences of discrete symbols. This makes the Ra-NEM approach more flexible, for example, applicable in natural language processing.

To train a Ra-NEM for supervised classifiers, we follow Petsiuk et al. (2018) and set $\delta_\Phi$ in (11) to the probability of the most likely class of the input image:

$$\delta_\Phi(x, x') = 1 - [\Phi(x')]_c \text{ with } c = \arg\max_i \Phi(x)_i \tag{12}$$

During training, for each input $x$, $\omega$ is chosen uniformly at random to be $\omega_{\text{ins}}$, $\omega_{\text{del}}$, or $\omega_{\text{rnd}}$ (see subsection 3.1).

## 4 EXPERIMENTS & RESULTS

To evaluate the Ra-NEM approach, we performed a series of experiments.

**Evaluated architectures and dataset.** We applied a diverse set of explanation methods to four deep learning models. Specifically, we studied three convolution-based architectures, ResNet50 (He et al., 2016), ConvNeXt (small) (Liu et al., 2022), and VGG16 (Simonyan and Zisserman, 2015), and one transformer architecture, ViT (Dosovitskiy et al., 2021). All models were sourced from the Timm library (Wightman, 2019) and pretrained on the ImageNet (Deng et al., 2009) training split. We focused on attribution methods for supervised image classifiers (Abhishek and Kamath, 2022). We ran evaluations using 1,000 images sampled from the ImageNet (Deng et al., 2009) validation split. For training NEMs, we sampled 10,000 images from the same split, discarding images that overlapped with the evaluation dataset.

**Baseline methods.** To compare our proposed methodology, we evaluated nine explanation methods comprising gradient-based and occlusion-based approaches. From the gradient-based methods, we considered Saliency (Simonyan and Zisserman, 2015), Grad-SHAP (Lundberg and Lee, 2017), Grad-CAM++ (Chattopadhay et al., 2018), and Integrated Gradients(Sundararajan et al., 2017). From occlusion-based methods, we examined RISE (Petsiuk et al., 2018), Smooth Pixel Mask (Fong et al., 2019), Information Bottleneck Attribution (IBA) (Schulz et al., 2020), and the recent NEMt (Møller et al., 2025). For IBA, we picked the *per-sample variant*, optimizing directly on the input image as recommended in the literature to achieve the highest faithfulness. We could not find any work on IBA for transformer-based image architectures and therefore did not generate results for explaining ViT with this method. For the Smooth Pixel Mask, we enforced an area constraint of 10%.

**NEM training and post processing.** We studied two different NEM architectures, Ra-NEM and NEMt. All NEM masking networks $\Psi$ were standard U-Net decoder blocks, consistent with previous work (Møller et al., 2024; 2025), to ensure a fair comparison. The trainable neural architectures used for the two methods were identical, which means that the number of trainable parameters was the same. All masking networks were trained for 10 epochs on the training data using the Prodigy optimizer (Mishchenko and Defazio, 2024) alongside a cosine annealing learn-rate scheduler. In Ra-NEM, we sampled six values of $k$ with replacement for each input $x$ in a batch and optimized for each sampled value, that is, we considered six loss terms 11 for each $x$. We set $\lambda = \frac{1}{50}$ in the NEMt loss function.

**Performance metrics.** We compared the different XAI methods using metrics for faithfulness, complexity, and inference time. Faithfulness was assessed using insertion and deletion curves, as is commonly done in the XAI literature (Petsiuk et al., 2018; Muzellec et al., 2024; Fong et al., 2019). We use the variant proposed by Muzellec et al. (2024). Specifically, for each attribution-image-model

combination, we compute the insertion and deletion curves and calculate the respective AUCs using $\omega_{\text{del}}(x, T)_i = 0$ and $\omega_{\text{ins}}(x, T)_i = \text{blur}(x)_i$ for $i \in T \subseteq [d]$. The final faithfulness metric is obtained by subtracting the deletion curve AUC from the insertion curve AUC. To further compare the methods, we measure their robustness to (unimportant) noise using the Average Sensitivity metric (Yeh et al., 2019). A common sanity check of attribution methods is to measure how much the attribution for the same input changes when the model is distorted by injecting random noise into the layers. We performed this check using a variant of the MPRT (Model Parameter Randomization Test, Adebayo et al. 2018) from the Quantus library (Hedström et al., 2023), where we measure the average spearman rank correlation between the attributions produced by the original model and a model with randomized weights. The full evaluation dataset ( 1000 images) was used to measure Faithfulness, whereas a subset of 100 images was used to measure Randomization and Robustness due to the metrics being computationally demanding. Finally, we measured the average time in seconds to generate attributions for the full evaluation dataset. Generally, low-latency methods are preferable, as real-time inference may be a requirement in practical applications. All experiments were run on a single RTX4090 Nvidia GPU.

**Results.** The results of our experiments comparing nine XAI methods are summarized in Table 1, the details are provided in Appendix B. All differences in Faithfulness between Ra-NEM and the other methods in Table 1 are highly statistically significant (two-sided paired Wilcoxon rank-sum test, $p < 0.001$). In our empirical evaluation, Ra-NEM achieved the highest average faithfulness. It gave the highest faithfulness for the ResNet50 ConvNeXt and ViT models, while it ranked fifth on the VGG16 model. With respect to Robustness, it ranks first overall indicating noise-resistant explanations. With a low Randomization score (0.051), Ra-NEM proves to be sensitive to the underlying model as desired and thus passes the sanity check. In terms of inference speed, Ra-NEM and NEMt were the fastest methods, with our implementation of Ra-NEM being the most efficient. Visual examples of the various attribution methods are shown in Figure 2 with additional images provided in Appendix F. Furthermore, we provide visual examples of Ra-NEM performing relatively poorly in Appendix G.

Table 1: Aggregated results from running different XAI methods on four different models using 1000 samples of the validation split of the ImageNet dataset. The explained models are a ConvNeXt, a ResNet50, a VGG16, and a ViT. We compared Faithfulness of Ra-NEM with the other methods, and the differences in the table below are statistically highly significant (two-sided paired Wilcoxon rank-sum test, $p < 0.001$). Results for each individual model are given in Appendix B. [†]IBA was only evaluated for the three CNN architectures. [‡]Grad-CAM++ was only evaluated for Robustness on the three CNN and NEMt was only evaluated on ViT and ResNet50. Please see Appendix B for further information.

| Method | Faith. ↑ | Robust. ↓ | Rand. [†][‡] | Time ↓ |
|---|---|---|---|---|
| RISE | 0.422 | 0.362 | -0.017 | 12.835 |
| Grad-CAM++[‡] | 0.390 | 0.599 | -0.082 | 0.013 |
| Integrated Gradients | 0.374 | 1.298 | 0.016 | 0.092 |
| Smooth Pixel Mask | 0.386 | 0.730 | 0.138 | 4.672 |
| Grad-SHAP | 0.343 | 1.350 | 0.015 | 0.014 |
| IBA[†] | 0.455 | 0.189 | -0.222 | 0.240 |
| Saliency | 0.294 | 1.332 | 0.075 | 0.008 |
| NEMt[‡] | 0.423 | 0.435 | 0.186 | 0.005 |
| Ra-NEM | **0.494** | **0.152** | 0.051 | **0.004** |

## 5 DISCUSSION

We derived a method for generating attributions by optimizing a Monte Carlo estimate of the AUC of the insertion/deletion curve via top-$k$ feature selection with uniformly sampled $k$ during training. Our experiments indicate that the proposed Ra-NEM algorithm is a robust choice to provide faithful attributions as measured by deletion and insertion curve AUC. It has a low latency that allows for real-time applications, is robust to small input perturbations and sensitive to the underlying model.

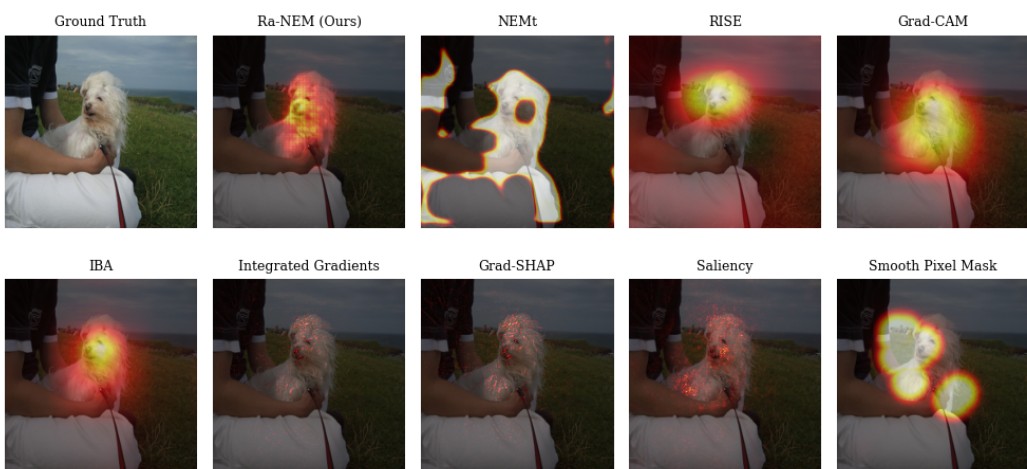

Figure 2: Model explanations generated by different XAI methodologies (see appendix for more examples). The explained model is a ResNet50 architecture trained on the training split of the ImageNet dataset. The example image is taken from the validation split of the ImageNet dataset. Compared to other methods, such as RISE and Grad-CAM++, Ra-NEM generates a ranking more closely aligned with the central object (dog), while the others exhibit a circular bias, likely due to smoothing. This may explain the higher faithfulness of Ra-NEM (see section 5).

**Faithfulness.** In our experiments, Ra-NEM achieved on average the highest faithfulness for all image classifiers. There was some variance in performance between models, but Ra-NEM had the highest faithfulness on three of four models. Ra-NEM gave the best result on the ConvNeXt model, where it improved 0.095 over NEMt ranking second. The worst relative performance of Ra-NEM was on VGG16, where it was 0.139 behind the first place taken by RISE. The strong average performance of Ra-NEM can be attributed to the inconsistent performance of other XAI methods when applied to different models, such as the poor performance of RISE on ViT and ConvNeXt. In contrast, our method gave faithful explanations for all models, even if it was outperformed for some individual models. To understand why our method performs well, it is therefore interesting to explore why the other methods vary in performance.Figure 2 exemplifies that the Ra-NEM attributions were more centered on the object (dog), while other occlusion-based methods have more circular smeared explanations. This is most likely due to smoothing, whether explicitly in the case of IBA (Schulz et al., 2020) and Smooth Pixel Mask (Fong et al., 2019) or implicitly by RISE (Petsiuk et al., 2018). This bias toward spatially cohesive masks can be good for some CNN-based methods such as VGG16 and ResNet50, but it might hamper the ability to explain methods using patching such as ViT and ConvNeXt. Previous work (Muzellec et al., 2024) has shown that gradient-based methods tend to exhibit reduced faithfulness when applied to CNN architectures due to the pooling operations. Thus, it could be argued that the good overall performance of Ra-NEM is partly due to not relying on spatial biases or gradients of the explained model. We hypothesize that the most important reason for Ra-NEM not achieving the highest faithfulness in all settings is that, unlike most other methods we compared against, Ra-NEM does not optimize directly on the output when generating attributions after training. Instead, it relies solely on prediction, and therefore can sometimes focus on the wrong areas as is evident in Appendix G.

**Randomization and robustness.** Our empirical results indicate that Ra-NEM is the most robust of the studied methods, where robustness against input noise is measured by Average Sensitivity. It even outperforms IBA and Grad-CAM++, which use smoothing to eliminate some of the noise inherent to gradient-based methods. Furthermore, Ra-NEM passed the randomization sanity check, since the MPRT score is close to zero (0.051). This means that the attribution depends on the underlying model mechanics instead of learning some shortcut such as segmenting the object in the image. Since Ra-NEM only has access to the latent representations of the input and not the input itself, this is not surprising.

**Time.** Ra-NEM was the fastest method for all models, closely followed by NEMt. This is in agreement with previous work on NEM methods (Møller et al., 2024; 2025), which outpaced other XAI methods when deployed. The NEM approaches only need a forward pass through the explained network $\Phi$ and the masking network $\Psi$, whereas the third fastest method Saliency needs to compute the gradient of the explained model. Since the $\Psi$ network is generally smaller than $\Phi$, a forward pass through both $\Psi$ and $\Phi$ is faster than a forward and backward pass through $\Phi$ (even when performed sequentially).

**Ra-NEM and previous NEM methods.** Compared to previous NEM algorithms (Møller et al., 2024; 2025), our method offers advantages beyond the reported improved performance results. Specifically, the Ra-NEM feature ranking and masking eliminate the need to specify a trade-off between complexity and accuracy. The binary masking facilitates adaptation to other input modalities, such as text or graphs, where partial occlusion is not well defined (see subsection 3.3). Additionally, the absence of tuning parameters in the loss function simplifies adoption, as there is no need to adjust hyperparameters when $\Phi$ is re-trained or the framework is applied in a new setting.

**Using the insertion/deletion curve as a target.** The insertion/deletion curves are widely considered as important measures of whether an attribution captures the underlying model mechanics in the XAI literature. Therefore, we studied the AUCs of these curves and an showed how to efficiently optimize them. It has been pointed out that feature removal metrics are sensitive to the choice of perturbation, how many perturbation steps are taken during feature removal, and whether the order of feature removal is ascending or descending in the attribution values (Tomsett et al., 2020). We mitigate these issues by considering both the insertion and deletion curves, which rely on different perturbation schemes. Furthermore, Ra-NEM uses three different perturbation schemes during training. In addition, Ra-NEM is optimized using a random $k$ during training, which means that it is not optimized with an apriori fixed number of steps. Our results show that these measures indeed lead to a high robustness of Ra-NEM, indicating that the results generalize. Finally, we would like to stress that optimizing the deletion/ insertion curve can be seen as a generalization of the objective of finding a good trade-off between complexity and accuracy, a common way of creating explanations (Møller et al., 2024; Fong et al., 2019; Muzellec et al., 2024), as the optimal insertion/deletion curve is one that has optimal accuracy for a given complexity level as defined by $k$.

**Limitations.** A potential limitation of our study is that Smooth Pixel Mask and NEMt belong to a different category of explanation methods than the others, which may introduce bias in direct comparisons (Møller et al., 2024; Fong et al., 2019). Specifically, these methods are classified as set-of-feature explanations, whereas the others fall under additive feature explanations (Møller et al., 2024). Set-of-feature explanations identify a minimal set of features sufficient to explain an outcome, but do not rank individual features. In contrast, additive feature explanations assign importance scores to all features, allowing for feature ranking. Comparing these approaches is inherently challenging, as they address different problems (Møller et al., 2024). We followed previous works (Muzellec et al., 2024; Petsiuk et al., 2018; Fong et al., 2019) in our evaluation strategy to allow for a direct comparison, but future work should explore whether these results generalize to other domains.

## 6 CONCLUSIONS

The areas under the insertion/deletion curves are often considered as important measures to evaluate the faithfulness of attribution-based XAI methods. This study contributes to a better conceptual understanding of this approach. We derived the sample complexity of a Monte Carlo approximation of the AUCs and an efficient differentiable objective function that allows one to optimize the AUC directly. These general findings can be used to improve XAI methods. We added the new objective function to the NEM approach, which leads to the Ra-NEM algorithm, which performs top-$k$ feature selection and optimizes faithfulness to produce more accurate attributions. In our experiments, Ra-NEM achieved state-of-the-art performance in terms of both faithfulness and robustness at low latency, enabling real-time processing. Applications with real-time constraints or that require generating numerous explanations, such as online video analysis, benefit not only from the high accuracy and low latency of Ra-NEM, but also from its resource efficiency. Ra-NEM extends the NEM framework and makes it applicable to a wider range of input modalities, which is an interesting direction for future work.

ETHICS STATEMENT

Advances in the field of explainable AI are crucial to improving fairness, transparency, trust, and accuracy in AI-driven decision-making.

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

## A  BOUNDS ON SAMPLE COMPLEXITY

The sample complexity of the AUC approximation can be bounded as follows:

**Theorem A.1.** *Given an input $x$ and an attribution $a$ with corresponding permutation $\sigma_a$, for randomly drawn indices $\mathcal{J}$ we have for $t > 0$*

$$P\big[|\widetilde{\mathrm{AUC}}_{ins}(x,a) - \mathrm{AUC}_{ins}(x,a)| > t\big] \leq 2e^{\frac{-2nt^2}{(1-f_n^*)(b-a)}} \quad , \tag{13}$$

*where $n = |\mathcal{J}|$; $a = \min \mathcal{C}$ and $b = \max \mathcal{C}$ for $\mathcal{C} = \{c_{ins}(i,x,a)\,|\,i = 1,\ldots,d\}$; and $f_n^* = (n-1)/d$.*

*Proof.* For a fixed input $x$ and an attribution $a$ with corresponding permutation $\sigma_a$, we have $\mathrm{AUC}_{\mathrm{ins}}(x,a) = \mu = \frac{1}{d}\sum_{c\in\mathcal{C}} c$. Setting $S_n = C_1 + \cdots + C_n$, where the $C_1,\ldots,C_n$ are $n$ samples drawn uniformly from $\mathcal{C}$ without replacement, we have $\widetilde{\mathrm{AUC}}_{\mathrm{ins}}(x,a) = S_n/n$. Now we apply a two-sided version of Corollary 1.1 from Serfling (1974) using $P_n = P[S_n/n - \mu \geq t]$ and $P_n = P[\mu - S_n/n - \mu \geq t]$, which completes the proof. □

Note that $\delta_\Phi$ does not need to be bounded for the proof. Setting $f_n^* = 0$ gives an expression that resembles what one would expect from a Hoeffding bound considering independent random variables. However, we cannot simply assume that the $c_{\mathrm{ins}}(i,x,a)$ terms in the AUC computation are independent. Anyway, the elements in $\mathcal{C}$ are fixed given an input $x$ and an attribution $a$ with corresponding permutation $\sigma_a$. Thus, we consider the problem of drawing without replacement from a set of numbers. We can also take the variance into account:

**Corollary A.2.** *With assumptions and definitions as in Theorem A.1 and its proof, we have*

$$P\big[|\widetilde{\mathrm{AUC}}_{ins}(x,a) - \mathrm{AUC}_{ins}(x,a)| > t\big] \leq 2\frac{(1-f_n)s^2}{nt^2} \quad , \tag{14}$$

*with $s^2 = \frac{1}{d}\sum_{c\in\mathcal{C}}(\mu - c)^2$ and $f_n^* = (n-1)/(d-1)$.*

*Proof.* The result follows from equation (1.4) by Serfling (1974). □

## B  DETAILED RESULTS

Here we report the main result tables for each individual explained model for a more granular insight into our experiments.

Table 2: Metrics from running nine different XAI methods on a ResNet50 architecture using 1000 samples of the validation split of the ImageNet dataset. We compared Faithfulness, Complexity and Sparsity results of Ra-NEM with the other methods, and the differences in the table below, excluding RISE when measuring faithfulness, are statistically highly significant (two-sided paired Wilcoxon rank-sum test, $p < 0.001$).

| Method | Faith. ↑ | Robust. ↓ | Rand. $^{\dagger}_{\dagger}$ | Time ↓ |
|---|---|---|---|---|
| RISE | 0.568 | 0.263 | -0.034 | 9.235 |
| Grad-CAM++ | 0.484 | 0.284 | 0.240 | 0.009 |
| Integrated Gradients | 0.397 | 1.360 | 0.017 | 0.046 |
| Smooth Pixel Mask | 0.436 | 0.729 | 0.032 | 3.349 |
| Grad-SHAP | 0.389 | 1.088 | 0.017 | 0.008 |
| IBA | 0.531 | **0.184** | -0.068 | 0.117 |
| Saliency | 0.396 | 0.893 | 0.096 | 0.007 |
| NEMt | 0.505 | 0.169 | 0.302 | 0.006 |
| Ra-NEM | **0.607** | 0.204 | 0.091 | **0.003** |

Table 3: Metrics from running nine different XAI methods on a ConvNeXt architecture using 1000 samples of the validation split of the ImageNet dataset. We compared Faithfulness, Complexity and Sparsity results of Ra-NEM with the other methods, and the differences in the table below are statistically highly significant (two-sided paired Wilcoxon rank-sum test, $p < 0.001$).

| Method | Faith. ↑ | Robust. ↓ | Rand. $^{\downarrow}_{\uparrow}$ | Time ↓ |
|---|---|---|---|---|
| RISE | 0.253 | 0.395 | -0.015 | 14.203 |
| Grad-CAM++ | 0.321 | **0.107** | -0.002 | 0.015 |
| Integrated Gradients | 0.332 | 1.756 | 0.012 | 0.081 |
| Smooth Pixel Mask | 0.251 | 0.744 | 0.275 | 5.026 |
| Grad-SHAP | 0.311 | 2.073 | 0.010 | 0.012 |
| IBA | 0.311 | 0.079 | -0.296 | 0.486 |
| Saliency | 0.158 | 2.667 | 0.008 | 0.009 |
| NEMt | 0.44 | 0.532 | 0.069 | 0.006 |
| Ra-NEM | **0.468** | 0.168 | -0.096 | **0.005** |

Table 4: Metrics from running nine different XAI methods on a VGG16 architecture using 1000 samples of the validation split of the ImageNet dataset. We compared Faithfulness, Complexity and Sparsity results of Ra-NEM with the other methods, and the differences in the table below, excluding NEMt when measuring faithfulness, are statistically highly significant (two-sided paired Wilcoxon rank-sum test, $p < 0.001$).

| Method | Faith. ↑ | Robust. ↓ | Rand. $^{\downarrow}_{\uparrow}$ | Time ↓ |
|---|---|---|---|---|
| RISE | **0.532** | 0.354 | -0.017 | 16.599 |
| Grad-CAM++ | 0.502 | 0.378 | -0.485 | 0.013 |
| Integrated Gradients | 0.313 | 0.975 | 0.017 | 0.115 |
| Smooth Pixel Mask | 0.474 | 0.670 | 0.033 | 3.873 |
| Grad-SHAP | 0.31 | 1.019 | 0.017 | 0.018 |
| IBA | 0.522 | 0.305 | -0.301 | 0.358 |
| Saliency | 0.296 | 0.923 | 0.053 | 0.006 |
| NEMt | 0.304 | 0.603 | n/a | 0.004 |
| Ra-NEM | 0.395 | **0.179** | 0.182 | **0.002** |

Table 5: Metrics from running nine different XAI methods on a ViT architecture using 1000 samples of the validation split of the ImageNet dataset. We compared Faithfulness, Complexity and Sparsity results of Ra-NEM with the other methods, and the differences in the table below, excluding Integrated Gradients when measuring faithfulness, are statistically highly significant (two-sided paired Wilcoxon rank-sum test, $p < 0.001$).

| Method | Faith. ↑ | Robust. ↓ | Rand. $^{\downarrow}_{\uparrow}$ | Time ↓ |
|---|---|---|---|---|
| RISE | 0.336 | 0.438 | -0.003 | 11.303 |
| Grad-CAM++ | 0.254 | 1.626 | n/a | 0.012 |
| Integrated Gradients | 0.456 | 1.104 | 0.017 | 0.124 |
| Smooth Pixel Mask | 0.384 | 0.777 | 0.213 | 6.441 |
| Grad-SHAP | 0.362 | 1.221 | 0.016 | 0.017 |
| IBA | n/a | n/a | n/a | n/a |
| Saliency | 0.325 | 0.846 | 0.143 | 0.009 |
| NEMt | 0.443 | 0.246 | 0.042 | 0.006 |
| Ra-NEM | **0.506** | **0.058** | 0.029 | **0.005** |

## C FORGRAD COMPARISON

For the CNN-based explained (frozen) models, the occlusion based XAI methods (e.g. RISE ) compared favourably to the gradient based approaches (e.g Integrated Gradients) in our experiments. An explanation for this can be found in the work by Muzellec et al. (2024), where it is shown that the poor performance of the gradient-based methods is likely due to the pooling operations leveraged in CNNs. The work furthermore introduce FORgrad, which is a method for improving the faithfullness of gradient based methods by filtering out high frequency noise in the gradient. In Table 6, we compare the FORgrad method applied to various Gradient based methods and compare them to Ra-NEM. It can be seen that the FORgrad method indeed improves the Faithfulness results but does not improve them to the quality of Ra-NEM.

Table 6: Comparing Faithfulness and Complexity metrics for Ra-NEM and gradient-based methods with and without FORgrad. We can see that FORgrad improves faithfullness at the cost of complexity. Ra-NEM is still outperforming the gradient-based methods even after they have been "repaired" by FORgrad.

| Method | Faithfulness ↑ |
|---|---|
| Grad-SHAP | 0.343 |
| Grad-SHAP (FORgrad) | 0.354 |
| Integrated Gradients | 0.374 |
| Integrated Gradients (FORgrad) | 0.407 |
| Saliency | 0.294 |
| Saliency (FORgrad) | 0.308 |
| Ra-NEM | **0.607** |

## D TRAINING NEM MODELS

In this section, we include various experimental results dealing with training Ra-NEM models. In Table 7, we explore the upfront training time needed for using Ra-NEM and compare it to NEMt. We can see that Ra-NEM does take longer to train, which is due to the extra samples used for training stability used by Ra-NEM.

Table 7: Time spend training NEM models for different explained models. The models are trained for 10 epochs on 10000 images extracted from the validation split of the ImageNet Model.

| Method | NEMt train time (sec.) ↓ | Ra-NEM train time (sec.) ↓ |
|---|---|---|
| ResNet50 | 335 | 700 |
| ConvNeXt | 360 | 869 |
| VGG16 | 437 | 980 |
| ViT | 403 | 913 |

Ra-NEM is a trained method, therefore its performance can potentially vary across training runs. To explore this, we trained a Ra-NEM to explain a ResNet50 multiple times and explored the variability in faithfulness across runs. The training protocol was identical to the one used for the main results. The experimental results can be seen in Table 8. We can see that performance was fairly stable across runs with a maximal difference in faithfulness of 0.021.

Table 8: Variability in faithfulness for a Ra-NEM explaining a ResNet50 across runs with same hyperparameters. The output appears to have high stability stable.

| Round | Faithfulness ↑ |
|---|---|
| 1 | 0.602 |
| 2 | 0.581 |
| 3 | 0.595 |
| 4 | 0.597 |
| 5 | 0.582 |

Finally, to understand the importance of choosing the number $k$ of samples used when training Ra-NEM, we trained a Ra-NEM to explain a ResNet50 model and varied the number of samples used throughout the runs. We subsequently measured training time and faithfulness to explore whether there exists a tradeoff between the two. The results are reported in Table 9, where it can be seen that there is some trade-off between training time and faithfulness, which is controlled via $k$.

Table 9: The effect of number of samples on faithfulness and training time for the Ra-NEM method when explaining a ResNet50 model. All hyperparameters except for number of samples are identical across runs. It can be seen that increasing $k$ can improve faithfulness but also increase training time.

| $k$ | Training time ↓ | Faithfullness ↑ |
|---|---|---|
| 1 | 252 | 0.520 |
| 2 | 310 | 0.560 |
| 4 | 463 | 0.596 |
| 6 | 700 | 0.607 |
| 8 | 967 | 0.617 |

# E  PSEUDOCODE FOR THE OMEGA OPERATOR

The pseudocode in PyTorch style in Algorithm 1 shows how we implemented $\omega(x, \mathrm{top}_k(a+g))$ in the loss function. The arguments A, x, k, and R refer to $a$, $x$, $k$, and the replacement values are denoted by $R \in \mathbb{R}^d$. We use the "Gumbel top-trick" for top-$k$ feature selection. We treat the attribution $a$ as logits. To ensure numerical stability, we normalize the attribution map using a logsumexp operation, which introduces a dependency between the individual feature attributions to prevent divergence. After the noisy top-$k$ selection, we map the attributions to $[0, 1]$ using a sigmoid centered on the $k$-th attribution value. After that, we apply a hard thresholding mapping the first $k$ attribution values to one and the others to zero. This step is ignored in the gradient computation using the straight-through operator (Bengio et al., 2013). This is why we perform soft thresholding using a sigmoid to guide the optimization. Essentially, we try to get the masked values close to either zero or one depending on it position relative to the $k$ largest feature value using the differentiable sigmoid function, before we perform the non-differentiable discretization step. Finally, the resulting mask is applied to the original input. The masked-out features are replaced depending on which $\omega$ is used, which is determined by the additional $R$ parameter in Algorithm 1. For example, $R = 0$ yields $\omega = \omega_{\mathrm{del}}$ and $R = \mathrm{blur}(x)$ corresponds to $\omega = \omega_{\mathrm{ins}}$.

# F  IMAGES FOR ALL MODELS

In this section, we show additional randomly selected example images for all models. We refer to the image captions for a qualitative discussion of the results.

```python
def omega_operator(A, x, k, R):
    # stabilize training
    A = A - A.logsumexp()
    # sample Gumbel noise
    U = torch.rand_like(A)
    G = -torch.log(-torch.log(U))
    # Gumbel perturbed sorting
    A = A + G
    A, indices = torch.sort(A, descending=True)
    # soft thresholding
    A = A - A[k]
    A = torch.sigmoid(A)
    # hard thresholding
    mask = A - A.detach()
    # only keep top k elements
    mask[:k] += 1
    # reorder mask to original index order
    M = torch.zeros_like(A)
    M[indices] = mask
    # generate perturbed input
    x_M = x * M + R * (1 - M)
    return x_M
```

**Algorithm 1:** Omega operator in Python/PyTorch pseudocode.

## F.1 RESNET50

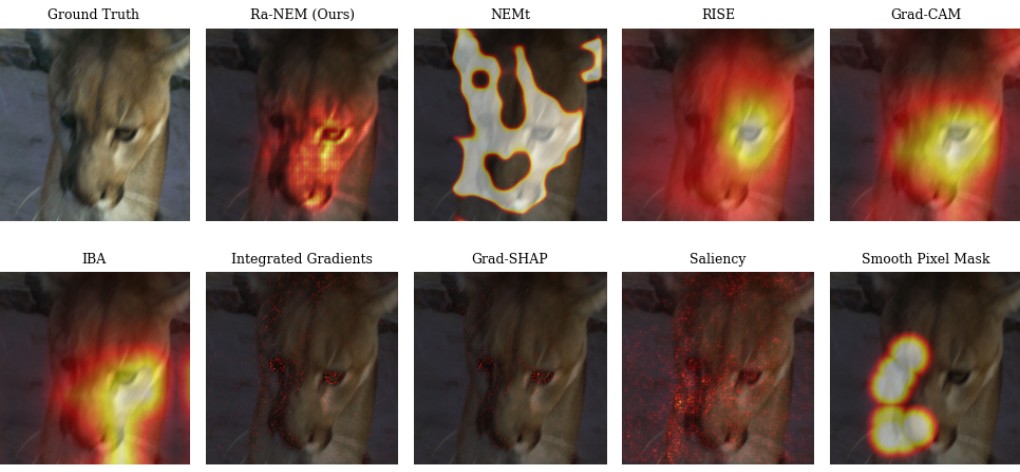

Figure 3: Model explanations generated by various different XAI methodologies. The explained model is a ResNet50 architecture trained on the training split of the ImageNet dataset. The example image is taken from the validation split of the ImageNet dataset. Comparing Ra-NEM with other occlusion-based methods, we observe that its finer granularity more precisely reveals specific image features. While all methods highlight the importance of the eye, Ra-NEM identifies only a small region of the eye and its outline as necessary.

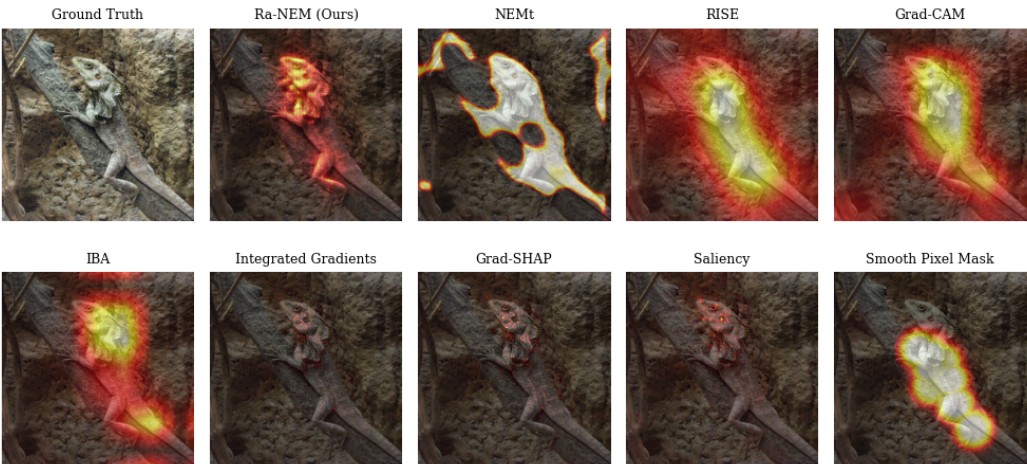

Figure 4: Model explanations generated by various different XAI methodologies. The explained model is a ResNet50 architecture trained on the training split of the ImageNet dataset. The example image is taken from the validation split of the ImageNet dataset.

## F.2 CONVNEXT

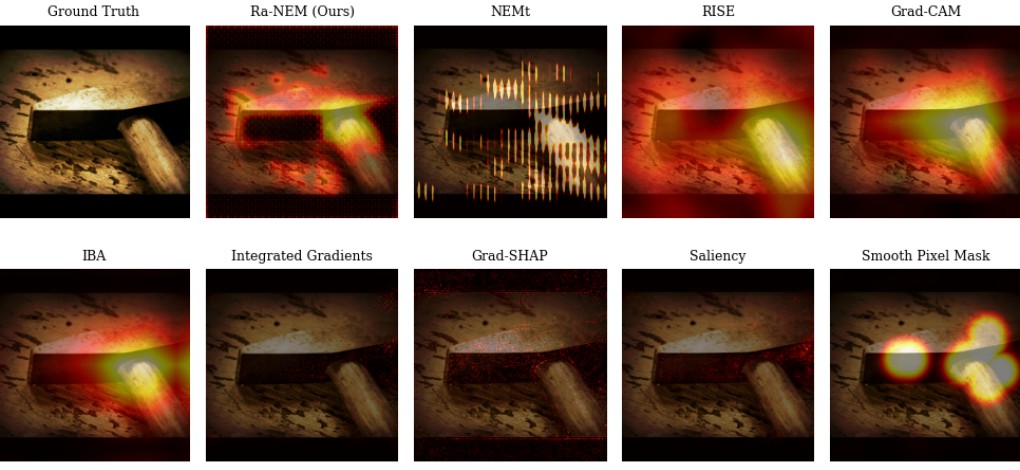

Figure 5: Model explanations generated by different XAI methodologies. The explained model is a ConvNeXt architecture trained on the training split of the ImageNet dataset. The example image is taken from the validation split of the ImageNet dataset. Occlusion-based methods generally agree on the area of interest, but only Ra-NEM—with its high granularity—can outline specific features. For example, it highlights both the shaft of the hammer and the outline of the hammerhead, which indicates that most of the hammerhead is not needed for classification.

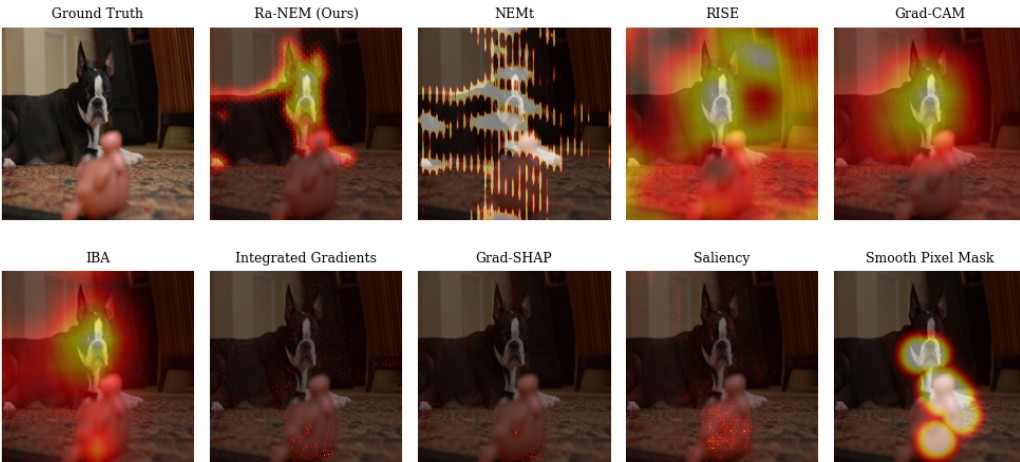

Figure 6: Model explanations generated by different XAI methodologies. The explained model is a ConvNeXt architecture trained on the training split of the ImageNet dataset. The example image is taken from the validation split of the ImageNet dataset. It can be seen that all occlusion-based methods agree that that the face of the dog is important, but Ra-NEM also emphasize the ears.

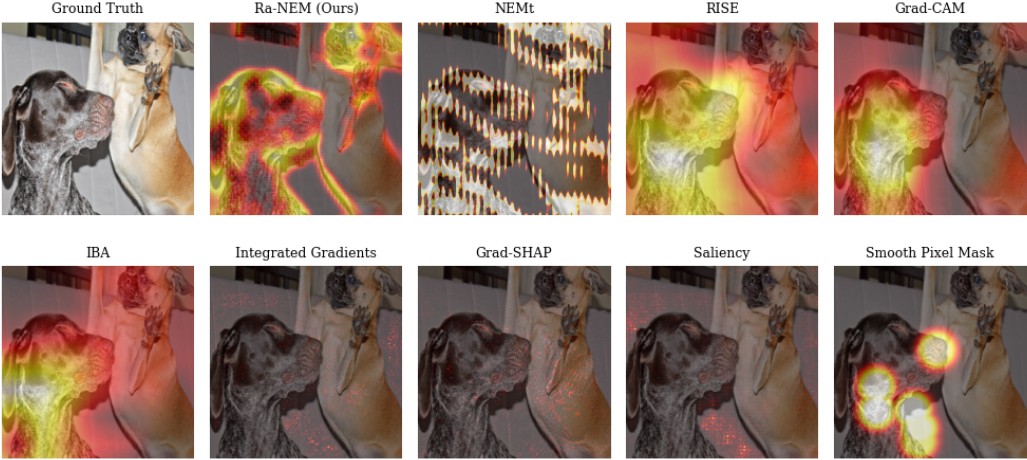

Figure 7: Model explanations generated by different XAI methodologies. The explained model is a ConvNeXt architecture trained on the training split of the ImageNet dataset. The example image is taken from the validation split of the ImageNet dataset. This example illustrates Ra-NEM's ability to discover fine details and as such indicate that the outline of the dog is central to classification, whereas other methods biased toward smooth explanations need to indicate the entire animal to be important.

## F.3 VGG16

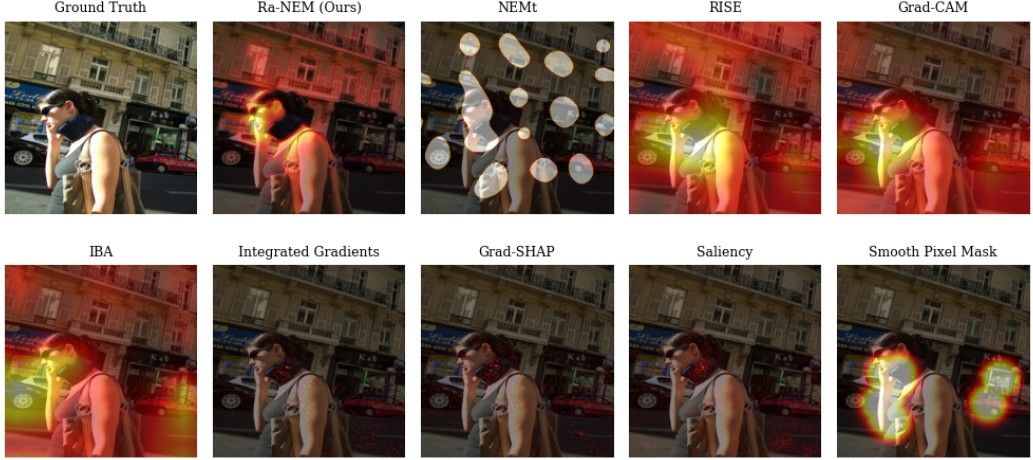

Figure 8: Model explanations generated by different XAI methodologies. The explained model is a VGG16 architecture trained on the training split of the ImageNet dataset. The example image is taken from the validation split of the ImageNet dataset. Notably, Ra-NEM excludes the neck brace, whereas the smoothing of the other occlusion based methods somewhat include it as an important feature. Furthermore, it can be seen that the gradient based methods all generally target the neck brace.

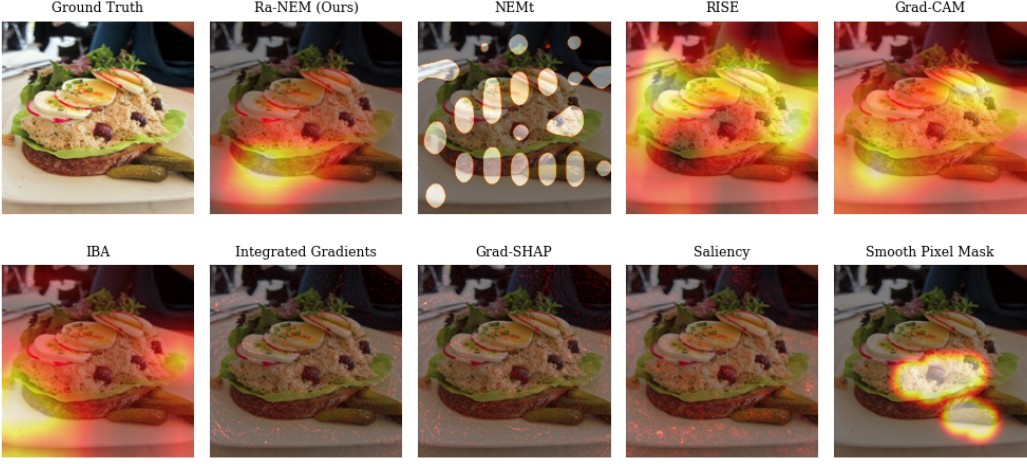

Figure 9: Model explanations generated by different XAI methodologies. The explained model is a VGG16 architecture trained on the training split of the ImageNet dataset. The example image is taken from the validation split of the ImageNet dataset.

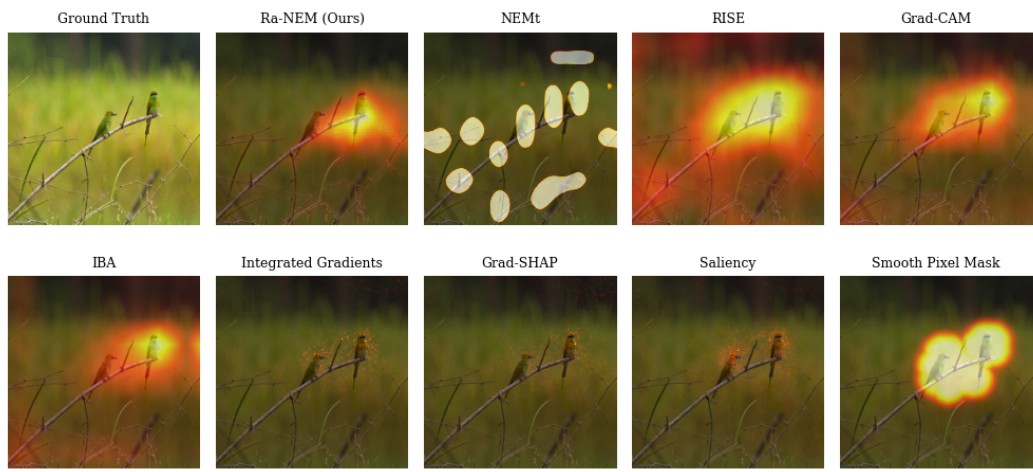

Figure 10: Model explanations generated by different XAI methodologies. The explained model is a VGG16 architecture trained on the training split of the ImageNet dataset. The example image is taken from the validation split of the ImageNet dataset.

## F.4 ViT

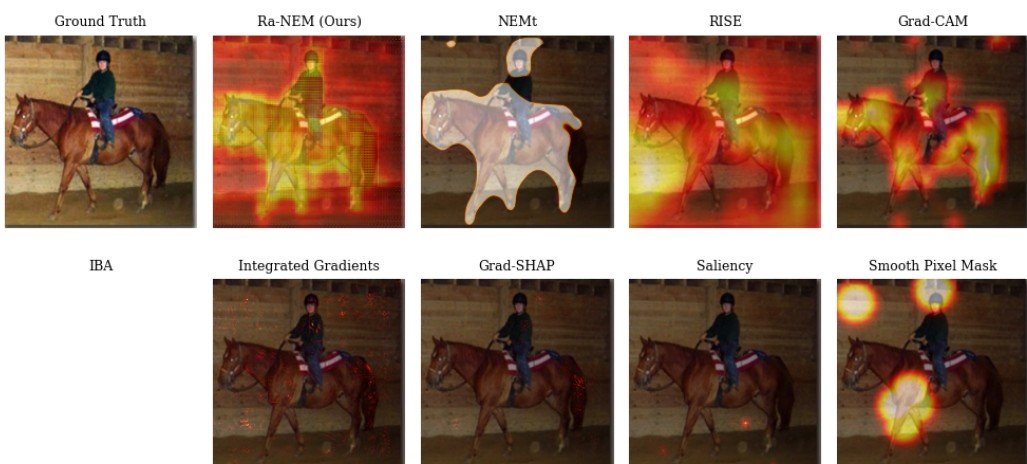

Figure 11: Model explanations generated by different XAI methodologies. The explained model is a ViT architecture trained on the training split of the ImageNet dataset. The example image is taken from the validation split of the ImageNet dataset. There is no adaptation of IBA for transformer-based image models, so IBA results have not been generated. This exemplary image illustrates a number of important points. We can see that the methods which leverage smoothing (Grad-CAM++ and RISE) generate attributions that are very scattered over the input, which might be due to an unfortunate combination of the smoothing and how the ViT processes an image. Given the ViT relies on attention mechanisms instead of convolutions, it could be that the architecture's bias toward spatial cohesion is much lower and therefore enforcing a spatial bias in the attributions might have an adverse effect. Additionally, while Integrated Gradients may achieve high faithfulness, its attributions offer limited interpretability for end users. In contrast, Ra-NEM provides a balanced trade-off between interpretability and faithfulness.

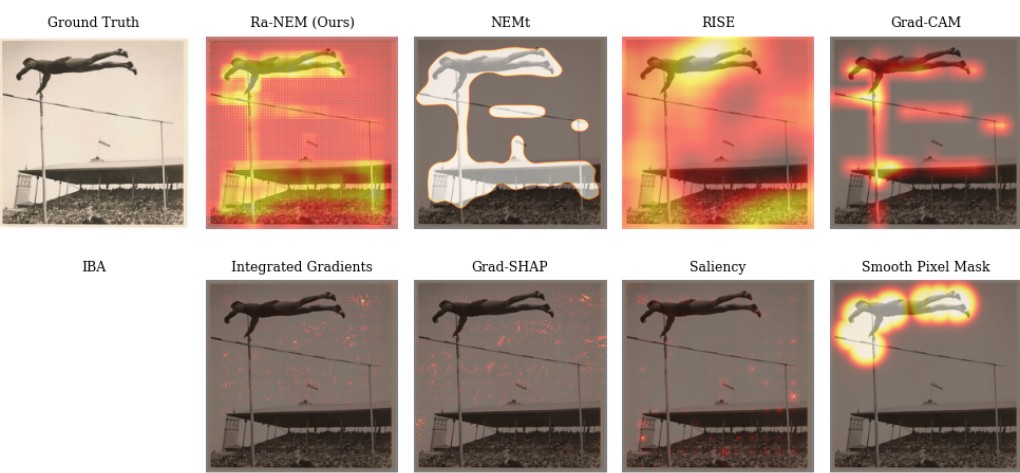

Figure 12: Model explanations generated by different XAI methodologies. The explained model is a ViT architecture trained on the training split of the ImageNet dataset. The example image is taken from the validation split of the ImageNet dataset. That there is no adaptation of IBA for transformer-based image models, so IBA results have not been generated.

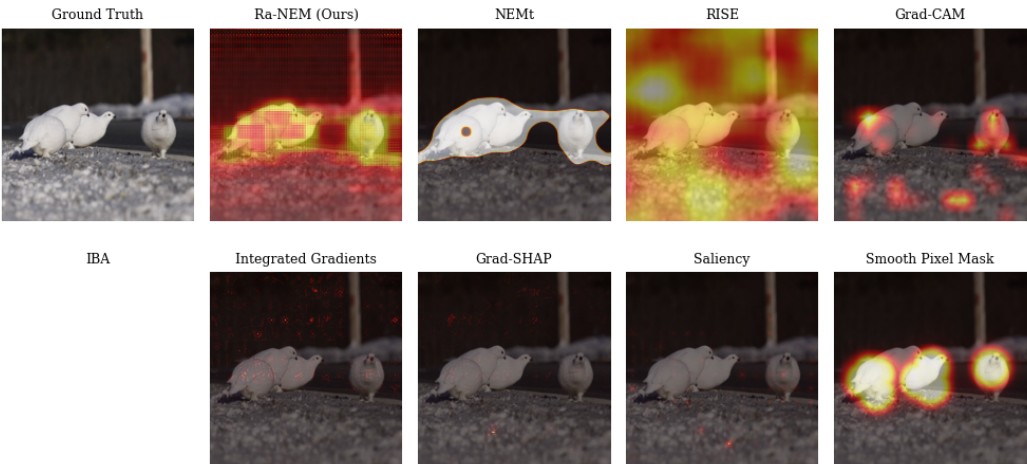

Figure 13: Model explanations generated by different XAI methodologies. The explained model is a ViT architecture trained on the training split of the ImageNet dataset. The example image is taken from the validation split of the ImageNet dataset. As there is no adaptation of IBA for transformer-based image models, IBA results have not been generated. This example underpins many of the points already discussed in Figure 11, a.i. the confusion of methods leveraging smoothing and the limited interpretability of Integrated Gradients, despite its higher faithfulness score. Additionally, NEMt occasionally generates very large attributions for the ViT.

# G    EXAMPLE IMAGES OF POOR RA-NEM PERFORMANCE

In the following, we show examples where Ra-NEM performed poorly compared to other methods to understand the limitations of the method. These examples were found by selecting the images for which the faithfulness scores of Ra-NEM were the lowest compared to the average faithfulness of all other methods.

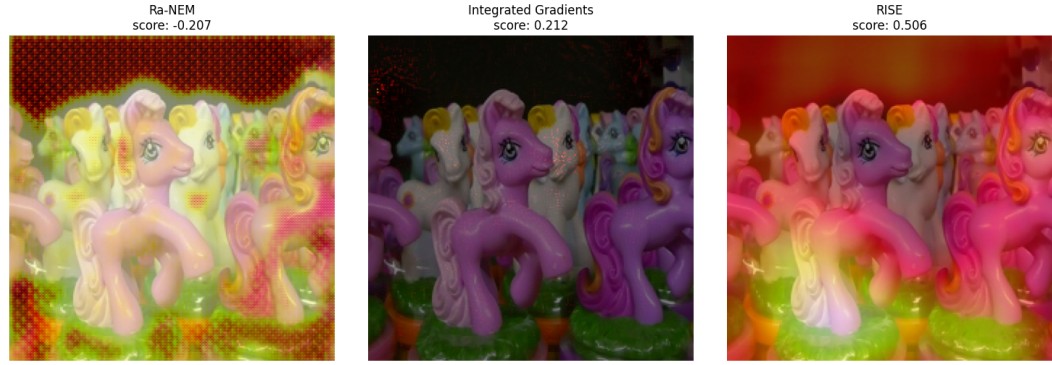

Figure 14: Example of image from ImageNet testset, where Ra-NEM performs relatively poor to other methods. Score is Faithfullness. We include Integrated Gradients and RISE for comparison. The explained model is a ConVNeXt trained on the trainings split of the ImageNet dataset.

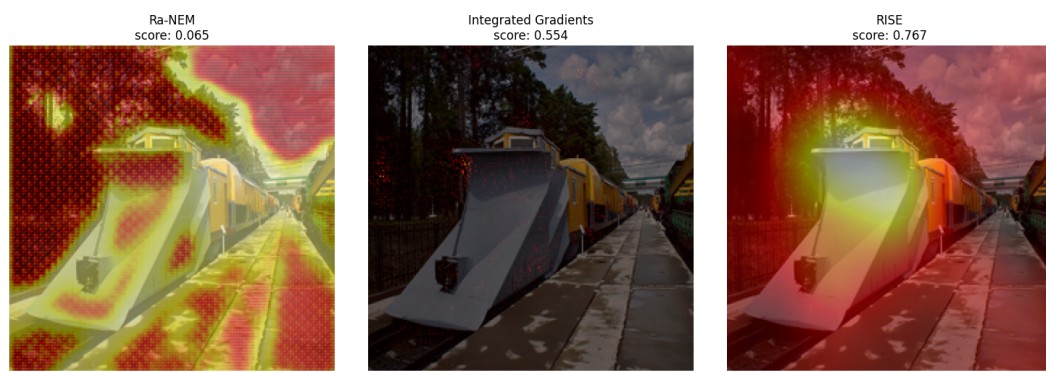

Figure 15: Example of image from ImageNet testset, where Ra-NEM performs relatively poor to other methods. Score is Faithfullness. We include Integrated Gradients and RISE for comparison. The explained model is a ConVNeXt trained on the trainings split of the ImageNet dataset.

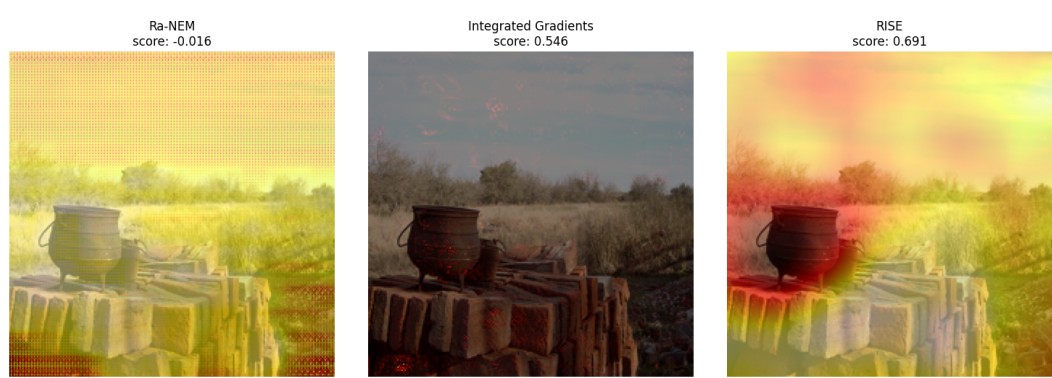

Figure 16: Example of image from ImageNet testset, where Ra-NEM performs relatively poor to other methods. Score is Faithfullness. We include Integrated Gradients and RISE for comparison. The explained model is a ViT trained on the trainings split of the ImageNet dataset. Here Ra-NEM appears to not properly distinguish between different elements in the images.

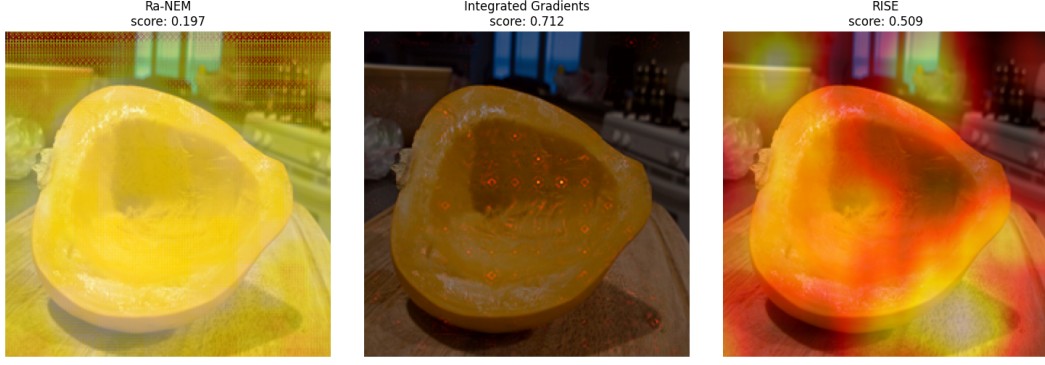

Figure 17: Example of image from ImageNet testset, where Ra-NEM performs relatively poor to other methods. Score is Faithfullness. We includeIntegrated Gradients and RISE for comparison. The explained model is a ViT trained on the trainings split of the ImageNet dataset. Here Ra-NEM again does not appear to properly distinguish between objects in the image.

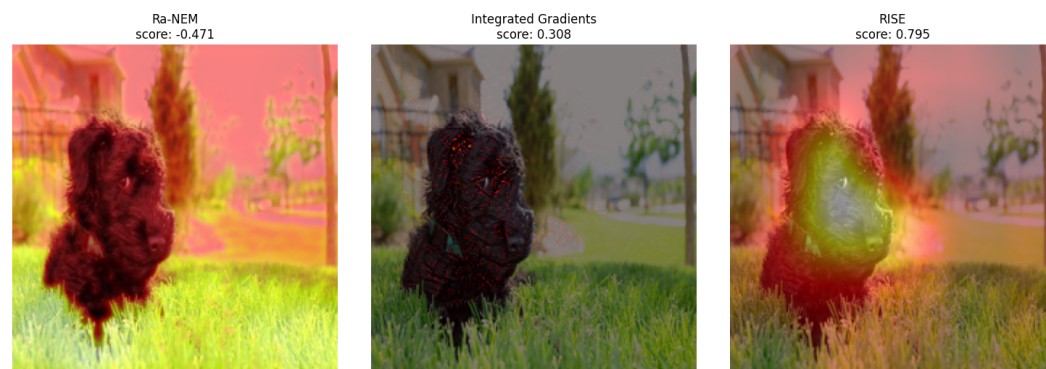

Figure 18: Example of image from ImageNet testset, where Ra-NEM performs relatively poor to other methods. Score is Faithfullness. We include Integrated Gradients and RISE for comparison. The explained model is a VGG16 trained on the trainings split of the ImageNet dataset. Here it can be seen, that Ra-NEM have switched foreground for background.

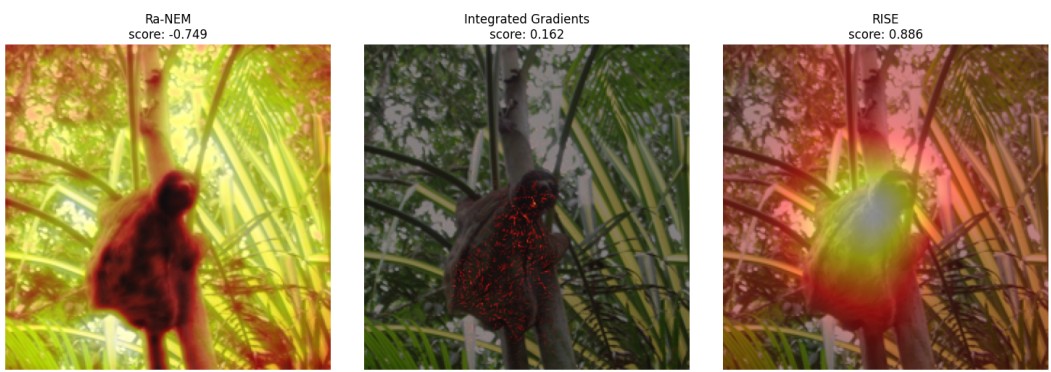

Figure 19: Example of image from ImageNet testset, where Ra-NEM performs relatively poor to other methods. Score is Faithfullness. We include Integrated Gradients and RISE for comparison. The explained model is a VGG16 trained on the trainings split of the ImageNet dataset. Here it can be seen, that Ra-NEM have again switched foreground for background.

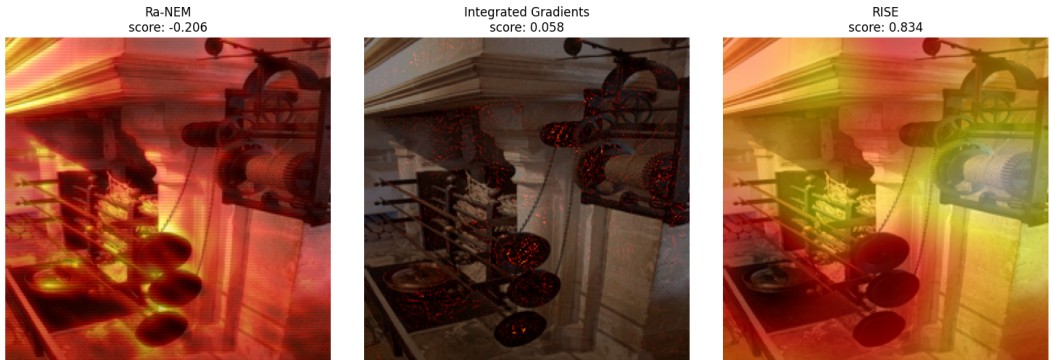

Figure 20: Example of image from ImageNet testset, where Ra-NEM performs relatively poor to other methods. Score is Faithfullness. We include Integrated Gradients and RISE for comparison. The explained model is a ResNet50 trained on the trainings split of the ImageNet dataset. Here Ra-NEM predicts that the wrong object is the most important.

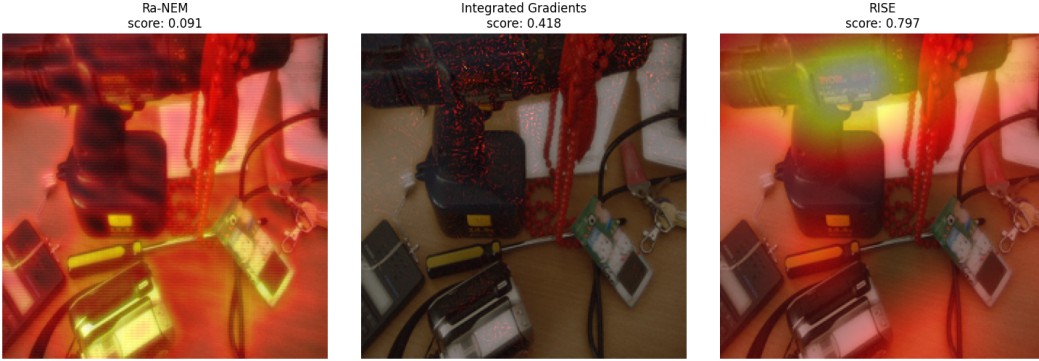

Figure 21: Example of image from ImageNet testset, where Ra-NEM performs relatively poor to other methods. Score is Faithfullness. We include Integrated Gradients and RISE for comparison. The explained model is a ResNet50 trained on the trainings split of the ImageNet dataset. Here Ra-NEM again focus on the wrong object in the image.

