# OpenReview forum: "Ra-NEM: Faithful model explanations through stochastic feature selection"
_ICLR.cc/2026/Conference — ICLR 2026 Conference Withdrawn Submission_

### Official Review · Reviewer_oTQK · 2025-10-28

**Soundness:** 2
**Presentation:** 2
**Contribution:** 1
**Rating:** 2
**Confidence:** 4

**Summary:**

The paper introduces Ra-NEM, an attribution method that aligns training with evaluation by directly optimizing the insertion and deletion AUC used to assess faithfulness. It reframes AUC maximization as a variable-k top-k selection problem, enabling a differentiable objective via Monte Carlo estimation and Gumbel top-k sampling. Integrated into a Neural Explanation Mask framework, the approach yields full-ranking, discrete masks with single-forward inference. Experiments across CNNs and ViTs report faithfulness, robustness, randomization sanity checks, and competitive runtime.

**Strengths:**

1. The paper is well organized and easy to follow, with a coherent narrative from problem framing to objective, method, and evaluation.

2. The combination of faithfulness, robustness, randomization sanity checks, and runtime offers a comprehensive assessment for the pixel insertion and deletion benchmarks. Results across multiple backbones also support the breadth of the claims.

3. The discussion provides detailed, thoughtful interpretation of results, which is encouraging.

**Weaknesses:**

1. Motivation conflates evaluation with objective. Attribution aims to faithfully explain model behavior. Insertion/deletion games are just one family of evaluation protocols, not the ground-truth objective of explanation. Prior work has also raised concerns about their reliability (e.g., input distribution shift). Therefore, treating insertion/deletion AUC as the optimization target risks optimizing the metric rather than faithfulness itself. The authors are suggested to reconsider the problem formulation and motivation.

2. Limited evidence beyond insertion/deletion benchmarks. While the insertion/deletion results are thorough, the paper provides insufficient support that the approach generalizes under alternative faithfulness criteria, perturbation models and other existing attribution benchmarks. A stronger case would include results on complementary benchmarks (e.g., infidelity tests, ROAR, pointing games, etc.) to demonstrate robustness to the choice of metric.

3. Incomplete computational cost accounting. Inference-time speed is reported, but training cost, which is part of the practical price of an explanation method, is not systematically quantified. The training time is also expected to be reported.

**Questions:**

1. What evidence supports that insertion/deletion AUC correlates with ground-truth explanatory quality rather than metric-specific artifacts?
2. How does the proposed method perform on other benchmark families?

---

### Official Review · Reviewer_4hzt · 2025-10-31

**Soundness:** 2
**Presentation:** 2
**Contribution:** 2
**Rating:** 4
**Confidence:** 4

**Summary:**

This paper introduces RA-NEM, a novel attribution method centered on a new, principled objective function: the direct optimization of the Area Under the Insertion Curve (AUC). The authors establish a theoretical link between maximizing the AUC and solving a top-k feature selection problem. To overcome the non-differentiable nature of this selection process, the "Gumbel top-k trick" is employed, allowing for end-to-end gradient-based training of an explanation-generating network. The paper provides empirical results on ImageNet benchmarks showing the method's superiority in faithfulness and efficiency.

**Strengths:**

1.  The primary strength of this paper is its move away from purely heuristic methods. By formulating a loss function that directly optimizes a widely accepted (though imperfect) evaluation metric, the work provides a more principled and less ad-hoc approach to generating explanations.
2.  Section 3.1 provides a clear and logical derivation, connecting the AUC calculation to a Monte Carlo approximation of top-k feature selection. The addition of Appendix A, which provides theoretical bounds on the sample complexity of this approximation, adds to the paper's theoretical grounding.
3.  The use of the Gumbel top-k trick is an elegant and appropriate solution for the non-differentiable sampling-without-replacement problem, which is more suitable than iterative Gumbel-Softmax applications. The implementation is clearly detailed in the pseudocode in Appendix E.

**Weaknesses:**

1.  The entire theoretical framework—that optimizing the AUC (a sum over all $k$) yields the optimal feature subset for *any* given $k$—relies on the "monotonicity" assumption stated in Section 3.1. This assumption (that $T_n^* \subset T_{n+1}^*$) is very strong and highly unlikely to hold true for complex deep models with significant feature interactions. The paper states this assumption but makes no effort to validate it or discuss the implications for the method if it is violated.
2.  The pseudocode in Algorithm 1 (Appendix E) reveals the use of a hard thresholding step (`mask - A.detach()`), which is a Straight-Through Estimator. This creates a known discrepancy between the forward pass (hard binary mask) and the backward pass (gradient from the sigmoid). STE can affect optimization stability and convergence, yet this is not analyzed or discussed as a limitation in the main paper.
3.  The training process involves two sources of randomness: the sampling of $k$ and the Gumbel noise. While Appendix D (Table 8) shows the *final* results are stable over 5 runs, the paper does not analyze how this variance impacts the *training dynamics* or the gradient quality during optimization.

**Questions:**

1.  Regarding the monotonicity assumption: Can the authors provide any empirical evidence or stronger justification for why this assumption should hold for architectures like ViT? If it does not hold, how does this affect the interpretation of what the RA-NEM loss is truly optimizing?
2.  Regarding Algorithm 1:
    a. What is the justification for the normalization step `A = A - A.logsumexp()` *before* adding Gumbel noise? The Gumbel top-k trick is typically applied directly to unnormalized logits.
    b. Can the authors provide an ablation study on the necessity of the STE? What happens to performance if the hard thresholding (lines 13-18) is removed and the model is trained to output the soft sigmoid mask directly?
3.  How sensitive is the method to the choice of $\delta_{\Phi}$? The method is trained to optimize the logit of the predicted class. Does this lead to "overfitting" to this specific metric, and would RA-NEM's superiority hold if faithfulness were evaluated using a different perturbation (e.g., mean-value imputation instead of blur) or a different output measure?

---

### Official Review · Reviewer_pqTb · 2025-10-31

**Soundness:** 2
**Presentation:** 2
**Contribution:** 2
**Rating:** 4
**Confidence:** 4

**Summary:**

This paper introduces RA-NEM, another entry in the crowded field of XAI attribution methods. It proposes to train an auxiliary network, based on the NEM framework, to generate explanations. Its main novelty is a loss function designed to directly optimize the insertion curve AUC, a common benchmark metric. The authors present experimental results on ImageNet, arguing that their method is more faithful, more robust, and significantly faster at inference time than nine other methods.

**Strengths:**

1.  The most compelling practical advantage is the inference speed. At 0.004s, it is orders of magnitude faster than high-performing occlusion methods like RISE (12.8s).
2.  I appreciate the authors' transparency in including an entire appendix (Appendix G) dedicated to the method's failure cases.

**Weaknesses:**

1.  The XAI field has many attribution methods. This work feels like a clever engineering exercise—combining the NEM framework with the Gumbel trick to optimize the AUC metric—with the primary goal of winning on a specific benchmark. It lacks a strong problem statement or motivating use case that demonstrates why existing SOTA methods are insufficient for a specific real-world problem.
2.  The paper's motivation claims to address "socially relevant tasks" and "real-world applications". However, the experiments are confined exclusively to ImageNet classification. There are no experiments on medical imaging, autonomous driving, or any other "socially relevant" domain. This is a significant disconnect between the paper's stated motivation and its execution.
3.  The paper's central premise is optimizing faithfulness(how well the explanation matches the model's logic). This is not the same as interpretability (how well a human can understand the model's decision). A "faithful" explanation of a model that relies on spurious background correlations (as seen in the failure cases in Appendix) is not interpretable or useful. Without a user study, claims of generating "useful explanations" are unsubstantiated.
4.  The "fast inference" claim is misleading in a practical sense. This method requires training, fine-tuning, and storing a separate, complex U-Net-based network for every single model one wishes to explain. This introduces a massive training and maintenance overhead in any practical MLOps pipeline, which is not required by post-hoc methods like Integrated Gradients or RISE. The training time, while a one-off cost, is non-trivial (700-980s per model).

**Questions:**

1.  The paper claims the binary masking approach makes RA-NEM "more flexible" and "applicable to a wider range of input modalities" like text. Why is there no supporting experiment, even a simple one, on text or tabular data? How is a U-Net architecture, which is heavily biased towards grid-like data, a suitable choice for non-grid modalities?
2.  Regarding the high training cost vs. high inference cost of methods like RISE: In a real-world setting where models are frequently retrained, this training cost must be paid every time. How do the authors justify this trade-off?
3.  Is the AUC optimization objective fundamentally tied to the NEM framework? Could this loss function be used in other contexts, for example, to directly optimize the mask in a method like Smooth Pixel Mask?

---

### Official Review · Reviewer_bJA9 · 2025-10-31

**Soundness:** 3
**Presentation:** 3
**Contribution:** 3
**Rating:** 6
**Confidence:** 3

**Summary:**

This paper proposes RA-NEM, a novel XAI attribution method that extends the Neural Explanation Mask (NEM) framework. The core idea is to generate high-faithfulness explanations by directly optimizing the Area Under the Insertion Curve (AUC), a common metric for explanation fidelity. The authors link this optimization problem to top-k feature selection and employ the Gumbel top-k trick to create a differentiable, end-to-end trainable system. The method is evaluated against nine baseline XAI methods on four different image classification models (ResNet50, ConvNeXt, VGG16, and ViT). The results demonstrate that RA-NEM achieves state-of-the-art or competitive performance in terms of faithfulness, robustness, and inference time.

**Strengths:**

1. The paper's primary claims are well-supported by the evidence. RA-NEM achieves the highest average Faithfulness (0.494) and Robustness (0.152) while also being the fastest method in terms of inference time (0.004s), as shown in Table 1. This combination is a significant achievement.
2.  The empirical evaluation is thorough, comparing RA-NEM against nine baselines across four diverse model architectures, including both CNNs and Transformers.
3.  The appendices are very detailed and proactively address many potential concerns. The inclusion of studies on training time, the impact of the $k$ parameter, and training stability adds significant rigor.
4.  The authors rightly include key sanity checks (MPRT for Randomization and Average Sensitivity for Robustness) directly in their main results table, which builds confidence in the method's reliability.

**Weaknesses:**

1. A notable weakness is the method's poor relative performance on the VGG16 model, where it ranks fifth in faithfulness, significantly behind RISE. While the authors offer a hypothesis (bias towards spatial cohesion in other methods), this undermines the claim of consistent SOTA performance across all architectures.
2.  A critical omission in an empirical paper. All results in Tables 1-5 are reported as mean values without standard deviations. Given the stochastic nature of the training (Gumbel noise, $k$ sampling) and the stability test in Table 8, reporting variance is essential for an accurate assessment of the results' significance.
3. The method produces highly sparse, fine-grained explanations (e.g., Figure 2, Figure 7), which score well on the faithfulness *metric*. However, it's not self-evident that these outline-like maps are more *interpretable* or useful to a human user than the smoother maps from methods like Grad-CAM. The paper equates high faithfulness with "useful explanations" without a user study to support it.

**Questions:**

1.  Regarding the poor VGG16 performance, could this indicate that the RA-NEM objective is less effective for older CNN architectures that might rely on different types of features than modern ConvNeXt or ViT models?
2.  The paper states a standard U-Net decoder was used. How sensitive is the final explanation's quality and sparsity to this specific decoder architecture? Have the authors explored simpler or different decoders?
3.  For the failure cases in Appendix G (e.g., Figure 18, 19, 20), are these "explanation failures" (i.e., RA-NEM failing to explain a correct classification) or are they faithfully reflecting a *model failure* (i.e., the original model was using background artifacts to make a correct classification)?

---

### Note · Authors · 2025-11-12

I have read and agree with the venue's withdrawal policy on behalf of myself and my co-authors.